# Association Between TikTok Use and Anxiety, Depression, and Sleepiness Among Adolescents: A Cross-Sectional Study in Greece

**DOI:** 10.3390/pediatric17020034

**Published:** 2025-03-11

**Authors:** Angeliki Bilali, Aglaia Katsiroumpa, Ioannis Koutelekos, Chrysoula Dafogianni, Parisis Gallos, Ioannis Moisoglou, Petros Galanis

**Affiliations:** 1P&A Kyriakou Children’s General Hospital, National and Kapodistrian University of Athens, 11527 Athens, Greece; aggelikimpi@nurs.uoa.gr; 2Clinical Epidemiology Laboratory, Faculty of Nursing, National and Kapodistrian University of Athens, 11527 Athens, Greece; aglaiakat@nurs.uoa.gr; 3Department of Nursing, University of West Attica, 12243 Athens, Greece; ikoutel@uniwa.gr (I.K.); cdafog@uniwa.gr (C.D.); parisgallos@nurs.uoa.gr (P.G.); 4Faculty of Nursing, University of Thessaly, 41500 Larissa, Greece; iomoysoglou@uth.gr

**Keywords:** TikTok, anxiety, depression, sleepiness, adolescents, problematic use, TikTok Addiction Scale

## Abstract

Introduction: TikTok use is increasing, especially among children and adolescents. However, the negative effects of TikTok use have not been sufficiently investigated. Aim: To examine the association between TikTok use and anxiety, depression, and sleepiness in adolescents. Methods: We conducted a web-based cross-sectional study in Greece. We employed a convenience sample of 219 adolescents in Greece. All participants had a TikTok account. We used the TikTok Addiction Scale (TTAS) to measure TikTok use and the Patient Health Questionnaire-4 (PHQ-4) to measure anxiety and depression. We constructed multivariable regression models, and we performed a stratified analysis according to gender. Results: The mean age was 18.5 years (standard deviation: 0.5, range: 18 to 19). In our sample, 81.3% were girls and 18.7% were boys. Mean daily TikTok usage and TTAS scores were statistically higher for girls than for boys. Our multivariable linear regression analysis was adjusted for age and showed that problematic TikTok use was associated with higher levels of anxiety and depression symptoms in both genders. In particular, we found a positive association between mood modification and anxiety score (beta = 0.404, *p*-value = 0.006). Moreover, we found a positive association between conflict and depression score (beta = 0.472, *p*-value < 0.001). Additionally, we found that the impact of TikTok on anxiety and depression was greater among boys than girls. We also found that problematic TikTok use was associated with increased sleepiness in both genders (mood modification: beta = 0.655, *p*-value < 0.001; conflict: beta = 0.674, *p*-value < 0.001). Conclusions: Our findings showed a positive association between TikTok use and anxiety, depression, and sleepiness. Early identification of problematic TikTok users is essential to promote their mental health and well-being. Healthcare professionals should be alert to recognize symptoms of problematic TikTok use.

## 1. Introduction

In the digital age, social media platforms have become essential in shaping human behavior and interactions with others. Among these platforms, TikTok has become a favorite among teenagers and young adults, with over 1 billion users worldwide [1]. Approximately 35% of TikTok users are between 16 and 24 years old [1]. TikTok is a short-form video-sharing app that allows users to consume and make short videos between 15 and 60 s in length, using filters, music, and lip-syncing templates. TikTok videos are algorithm-driven and tailored to users’ indicated preferences and previously liked content [2]. Thus, the immersive design of TikTok may cause compulsive use, especially among adolescents, since they are particularly vulnerable to the effects of social media because adolescence is a critical developmental period characterized by intense physical, emotional, and social changes [3].

We should recognize that technology, including social media, may be beneficial for users. Several reviews have shown that video games can be a valuable tool for the cognitive and social development of children and adolescents since video games have positive effects on cognitive skills, enhanced hand-eye coordination, reaction time, creativity, problem-solving, spatial abilities, teamwork, and communication [4,5,6]. In brief, we can organize the positive effects of playing video games into four domains: social, emotional, motivational, and cognitive [5]. For instance, D’Errico et al. found a positive association between playing video games and protective behavioral intentions, internal locus of control, risk perception, and engagement [7]. Moreover, Greitemeyer found that exposure to prosocial models through video games could be an effective way to promote positive social encounters [8].

Several systematic reviews have shown the negative influence of Facebook, Instagram, and Twitter (X), or social media in general, without including TikTok, on adolescents’ mental health [3,9,10]. For instance, Keles et al. found that time spent on social media, social media activity, investment, and addiction were correlated with psychological distress, anxiety, and depression [9]. Similarly, Saleem et al. identified 67 studies that showed a positive relationship between problematic social media use and anxiety and depression [10]. Moreover, Kerr et al. included 32 studies in their systematic review and found a positive association between problematic social media use and anxiety symptoms [3].

Although the association between problematic Facebook, Instagram, and Twitter (X) use and mental health issues among adolescents is well established, little is known regarding the association between TikTok use and anxiety, depression, and sleepiness among adolescents. To the best of our knowledge, only four studies have examined the association between TikTok use and anxiety, depression, and sleep quality among adolescents [11,12,13,14]. In particular, two studies examined the association between TikTok use and anxiety symptoms [11,13], four studies investigated the association between TikTok use and depressive symptoms [11,12,13,14], and one study explored the association between TikTok use and sleep quality [11]. Chao et al. examined 1346 adolescents (mean age: 14.5 years) in China and found that TikTok-addicted users show worse mental health conditions, such as higher levels of stress, anxiety, sadness, loneliness, social anxiety, focus issues, and poorer sleep and life satisfaction compared to non-users and moderate users. Additionally, TikTok-addicted users experienced inferior academic performance, more bullying victimization, worse family connections, and higher academic stress [11]. Sha and Dong included 3036 Chinese students in their study, with a mean age of 16.7 years, and found a positive relationship between TikTok use disorder and anxiety, depression, and stress [13]. Also, Gentzler et al. employed a sample of 237 American adolescents (mean age: 15.1 years) and found a positive relationship between total time spent on TikTok and depressive symptoms [12]. Ilic-Zivojinovic et al. investigated 620 students (mean age: 16.1 years) in Serbia and found that depressive symptoms are more frequent among TikTok users than among non-users [14]. We should note that Gentzler et al. [12] only measured total time spent on TikTok, without using a valid scale to measure levels of TikTok use. Also, Ilic-Zivojinovic et al. did not measure the frequency of TikTok use but divided their participants into TikTok users and non-users based on a yes/no question. Moreover, Chao et al. [11] and Sha and Dong [13] used a valid scale to measure TikTok use (i.e., Smartphone Addiction Scale (SAS) [15]); however, the SAS is not a specific scale for measuring TikTok use since it was constructed to measure smartphone addiction among users. Chao et al. [11] and Sha and Dong [13] used the SAS by changing the word “smartphone” to the word “TikTok” throughout the scale.

Several other studies have shown a negative association between TikTok use and mental health issues among adolescents beyond anxiety, depression, and sleep quality. In brief, the literature suggests a negative impact of TikTok on self-body image, satisfaction with physical appearance, eating disorders, social isolation, unhappiness, self-esteem, concentration, time distortion, and anger [16,17,18,19,20,21,22,23,24]. All these studies measured only the total time spent on TikTok without using a valid scale.

We should note that the literature regarding the association between TikTok use and mental health issues is more extensive for adults than for adolescents. In brief, several studies involving adults have shown a positive association between problematic TikTok use and anxiety [25,26,27,28], depression [25,27,28,29,30,31], poor sleep [32], stress [27], disorder eating behavior [18,33], and loneliness [29,31]. Moreover, a recent meta-analysis found that problematic TikTok use is associated with anxiety (adjusted beta = 0.406, 95% confidence interval: 0.279 to 0.533, *p* < 0.001) and depression (adjusted beta = 0.321, 95% confidence interval: 0.261 to 0.381, *p* < 0.001) [34].

As mentioned above in detail, there are only four studies that examined the association between TikTok use, anxiety, depression, and sleep quality among adolescents without using a valid scale to measure TikTok use. In this context, we performed a study to examine the association between TikTok use and anxiety, depression, and sleepiness in adolescents since the literature in this field is scarce. We investigated this association for the first time in a European country, i.e., Greece, by using a specific tool to measure problematic TikTok use. Moreover, to the best of our knowledge, this is the first study that used a valid scale to measure problematic TikTok use, namely the TikTok Addiction Scale. Additionally, we examined for the first time the potential role of gender as a moderator in the relationship between TikTok use, anxiety, depression, and sleepiness. Thus, our research questions are as follows:Is there an association between TikTok use and anxiety in adolescents? We expected a positive association between variables.Is there an association between TikTok use and depression in adolescents? We expected a positive association between variables.Is there an association between TikTok use and sleepiness in adolescents? We expected a positive association between variables.Are there possible differences between girls and boys regarding the association between TikTok use, anxiety, depression, and sleepiness in adolescents? We expected differences between the two genders.

## 2. Materials and Methods

### 2.1. Study Design

The World Health Organization defines adolescence as the phase of life between childhood and adulthood, from ages 10 to 19 [35]. In this context, we conducted a cross-sectional study in Greece, including adolescents aged 18 and 19 years. We developed an online version of the study questionnaire using Google Forms and disseminated it through TikTok. In particular, we created a TikTok video that informs TikTok users about our study. Users who want to participate in our study must be students between 18 and 19 years old. Moreover, our participants should understand the Greek language. Among TikTok users, 264 expressed interest in participating in our study. We sent the Google form link to TikTok users who wanted to participate in our study through an inbox message. A total of 219 TikTok users completed the study questionnaire. Therefore, the response rate was 83.0% (219 out of 264). Before initiating the online survey, we presented participants with an introductory page containing essential information. This page outlined the study’s purpose and structure, briefly explained the questions participants would be asked to complete, estimated the time required to finish the questionnaire, emphasized the voluntary nature of participation, and informed participants of their ability to exit the survey by closing their web browser. Additionally, we included the researcher’s contact information. To ensure data integrity, we inquired whether participants had previously completed the survey, and any affirmative responses were subsequently removed from the dataset. Thus, we obtained a convenience sample. We collected our data in December 2024.

We used G*Power v.3.1.9.2 to calculate our sample size. Considering a small effect size between TikTok problematic use and anxiety, depression, and sleepiness (f^2^ = 0.07), the number of independent variables (six predictors and one confounder), a confidence level of 95%, and a margin of error of 5%, the sample size was estimated at 188 participants.

### 2.2. Measurements

We measured the gender and age of our participants. We asked participants about their sex assigned at birth. We measured the time of TikTok use with a simple question: “How many hours do you use TikTok typically in a day?”.

Sleepiness was measured with a single item: “How sleepy do you feel in class?”. Answers were recorded on a 10-point Likert scale, ranging from 1 (not sleepy at all) to 10 (very sleepy). Several valid scales use a single item to measure subjective levels of sleepiness, such as the Karolinska Sleepiness Scale [36] and the Stanford Sleepiness Scale [37]. The Karolinska Sleepiness Scale measures individuals’ sleepiness during the five minutes before the rating, and answers are recorded on a 10-point Likert scale, ranging from 1 (extremely alert) to 10 (extremely sleepy) [36]. Moreover, the Stanford Sleepiness Scale measures the level of perceived sleepiness at a specific moment in time, and answers are recorded on a 7-point Likert scale, ranging from 1 (feeling alert) to 7 (lost the struggle to remain awake) [37]. In this context, we used a 10-point Likert scale ranging from 1 (not sleepy at all) to 10 (very sleepy) to subjectively assess the level of sleepiness in our participants. In other words, our 10-point Likert scale was a slight adaptation of valid single-item scales, such as the Karolinska Sleepiness Scale [36] and the Stanford Sleepiness Scale [37].

We measured problematic TikTok use using the Greek version of the TikTok Addiction Scale (TTAS) [38]. The TTAS includes fifteen items and measures six factors: salience (two items), mood modification (two items), tolerance (three items), withdrawal symptoms (two items), conflict (four items), and relapse (two items). The concept of salience in TikTok usage refers to users’ fixation on the platform, while mood modification describes its capacity to enhance one’s emotional state. Tolerance is observed when users require increased engagement with TikTok to achieve satisfaction, and withdrawal manifests as negative emotions when usage is stopped. Conflict arises when TikTok interferes with daily activities, and relapse occurs when users return to previous usage patterns after a period of abstinence. The TTAS measures individuals’ attitudes towards TikTok over the last 12 months. Answers are recorded on a five-point Likert scale, ranging from 1 (very rarely) to 5 (very often). Total scores and scores on factors range from 1 to 5. Higher scores indicate higher levels of problematic TikTok use. There is a suggested cut-off point score of 3.23 to distinguish healthy users from problematic TikTok users [39]. We used the valid Greek version of the TTAS [38]. In our study, Cronbach’s alpha for the TTAS was 0.905, while Cronbach’s alpha for the six factors ranged from 0.652 to 0.869.

We measured anxiety and depression using the Patient Health Questionnaire-4 (PHQ-4) [40]. The PHQ-4 includes four items and measures anxiety (two items) and depression (two items). Answers are provided on a four-point Likert scale, ranging from 0 (not at all) to 3 (nearly every day). Scores on the two factors range from 0 to 6, with higher scores indicating higher levels of anxiety and depressive symptoms. Scores of 3 or greater are considered positive for screening purposes. We used the Greek version of the PHQ-4 [41]. In our study, Cronbach’s alpha was 0.803 for the PHQ-4, 0.823 for anxiety, and 0.704 for depression.

### 2.3. Ethical Issues

Our study protocol was approved by the Ethics Committee of the Faculty of Nursing, National and Kapodistrian University of Athens (approval number; 510, June 2024). Moreover, we followed the guidelines of the Declaration of Helsinki to perform our study [42]. We informed our participants about the study design, and we asked whether they consented to participate in our study. In particular, before TikTok users were allowed to have access to the online study questionnaire, they were asked through Google Forms if they agreed to participate in our study. Participants with a positive answer could then fill in the study questionnaire. In this way, we obtained informed consent. Moreover, we did not collect personal data from the participants. Therefore, participation was voluntary and anonymous. Since our participants were aged between 18 and 19 years, there was no need to obtain informed consent from their parents.

### 2.4. Statistical Analysis

We present categorical variables with numbers and percentages. Additionally, we present continuous variables with the mean, standard deviation (SD), median, minimum value, and maximum value. Also, we present skewness and kurtosis values. We considered the six factors of the TTAS as independent variables. However, the tolerance and variance inflation factor indices for the factors “salience”, “tolerance”, and “withdrawal” had unacceptable values (tolerance lower than 0.5 and variance inflation factor higher than 4) [43]. Therefore, to avoid multicollinearity issues in the multivariable regression models, we had to remove three factors (i.e., salience, tolerance, and withdrawal) from the final models. We considered anxiety score, depression score, and sleepiness as dependent variables. We used the Kolmogorov–Smirnov test and Q-Q plots to assess the distribution of the continuous variables. Since the dependent variables are continuous variables that followed a normal distribution, we applied linear regression analysis. In this case, we present unadjusted and adjusted coefficients beta, 95% confidence intervals (CI), and *p*-values. We adjusted all models for age. We used variance inflation factors (VIFs) to assess multicollinearity in the multivariable models. A VIF greater than 5 indicates multicollinearity between independent variables [44]. VIFs for the final multivariable models ranged from 1.006 to 1.620; thus, there were no multicollinearity issues. We examined histograms of the residuals to check for multivariable normality. We examined scatterplots of residuals versus predicted values to check for homoscedasticity and linearity. We performed a stratified analysis according to gender to identify possible differences between girls and boys. We compared anxiety, depression, daily TikTok usage, and TTAS scores between the two genders using independent samples *t*-tests. Also, we used a chi-squared test to compare the percentage of problematic TikTok use among the two genders. *p*-values less than 0.05 were considered statistically significant. We used IBM SPSS 28.0 (IBM Corp. Released 2021. IBM SPSS Statistics for Windows, Version 28.0. Armonk, NY, USA: IBM Corp) for the analysis.

## 3. Results

### 3.1. Demographic Characteristics

The study population included 219 participants. Most of them were girls (81.3%, n = 178), while 18.7% (n = 41) were boys. Moreover, 51.1% (n = 112) were 19 years old, and 48.9% (n = 107) were 18 years old. The mean sleepiness score was 5.84 (SD: 2.09), with a median value of 6 (range: 0 to 10, skewness: −0.42, kurtosis: −0.21).

### 3.2. TikTok Use

The mean daily TikTok usage was 2.77 h (SD: 1.79), with a median of 2 h (range: 0.1 to 8). Mean daily TikTok usage was statistically higher for girls than boys (2.87 vs. 2.32, *t*-test = 2.32, df = 89, mean difference = 0.55, 95% CI = 0.08 to 1.03, Cohen’s d = 0.35, *p*-value = 0.023). The mean TTAS score was 2.48 (SD: 0.69), and the median score was also 2.47 (range: 1.07 to 4.07). The mean TTAS score was statistically higher for girls than boys (2.53 vs. 2.27, *t*-test = 2.23, df = 217, mean difference = 0.26, 95% CI = 0.03 to 0.50, Cohen’s d = 0.39, *p*-value = 0.027). Table 1 presents descriptive statistics for TTAS. The factors “mood modification” and “tolerance” had the highest mean scores, followed by “conflict” and “salience”. The factors “relapse” and “withdrawal symptoms” showed the lowest mean scores. Among the participants, 16.9% (n = 37) had a mean score ≥ 3.23, indicating problematic TikTok use, while 83.1% had a mean score < 3.23, suggesting healthy usage patterns. The prevalence of TikTok problematic use was higher among girls (19.1%, n = 34) than boys (7.3%, n = 3), but the difference was not statistically significant (chi-squared test = 3.30; df = 1; Phi = 0.12; odds ratio = 2.99; 95% CI = 0.87 to 10.27; *p*-value = 0.07).

### 3.3. Anxiety and Depression

The mean anxiety score was 2.95 (SD: 1.74) with a median value of 3 (range: 0 to 6, skewness: 0.28, kurtosis: −0.85), while the mean depression score was 2.11 (SD: 1.57) with a median value of 2 (range: 0 to 6, skewness: 0.94, kurtosis: 0.37). More than half of our participants (55.7%, n = 122) had an anxiety score ≥3, indicating considerable anxiety issues. Moreover, 31.5% (n = 69) had a depression score of 3 or greater, indicating high levels of depressive symptoms. We did not find differences in anxiety (*p*-value = 0.590) and depression (*p*-value = 0.767) scores between the two genders. In particular, the mean anxiety score for boys and girls was 2.80 (SD: 1.94) and 2.98 (SD: 1.70), respectively. Moreover, the mean depression score for boys and girls was 2.17 (SD: 1.83) and 2.09 (SD: 1.51), respectively.

### 3.4. Association Between TikTok Use and Anxiety

Bivariate analysis showed that the mean anxiety score for girls was 2.98 (SD: 1.70), and for boys it was 2.80 (SD: 1.94) (*t*-test = 0.54, df = 55, mean difference = 0.18, 95% CI = −0.48 to 0.84, Cohen’s d = 0.10, *p*-value = 0.59). Pearson’s correlation coefficients between anxiety and mood modification, conflict, and relapse were 0.25 (*p*-value < 0.001), 0.15 (*p*-value = 0.03), and 0.15 (*p*-value = 0.02), respectively.

Multivariable analysis of the full sample showed a positive association between mood modification and anxiety score (adjusted coefficient beta = 0.404, 95% CI = 0.115 to 0.693, *p*-value = 0.006). Stratified analysis showed that the impact of mood modification on anxiety score was higher among boys (adjusted coefficient beta = 0.760, 95% CI = 0.121 to 1.399, *p*-value = 0.021) than among girls (adjusted coefficient beta = 0.338, 95% CI = 0.031 to 0.645, *p*-value = 0.031). Moreover, we found a statistically significant and positive association between conflict and anxiety score among boys (adjusted coefficient beta = 1.236, 95% CI = 0.763 to 1.710, *p*-value < 0.006), but this association did not exist among girls. Table 2 shows univariate and multivariable linear regression models with anxiety score as the dependent variable. Figure 1 indicates multivariable normality since the residuals followed a normal distribution. Figure 2 indicates the homoscedasticity and linearity of the multivariable model, with anxiety score as the dependent variable.

### 3.5. Association Between TikTok Use and Depression

Bivariate analysis showed that the mean depression score for girls was 2.09 (SD: 1.51), and for boys it was 2.17 (SD: 1.83) (*t*-test = 0.29, df = 217, mean difference = 0.08, 95% CI = −0.62 to 0.46, Cohen’s d = 0.05, *p*-value = 0.77). Pearson’s correlation coefficients between depression and mood modification, conflict, and relapse were 0.17 (*p*-value = 0.01), 0.34 (*p*-value < 0.001), and 0.23 (*p*-value = 0.001), respectively.

Multivariable analysis of the full sample showed a positive association between conflict and depression score (adjusted coefficient beta = 0.472, 95% CI = 0.239 to 0.704, *p*-value < 0.001). This association was consistent in both boys and girls. Moreover, stratified analysis showed that the impact of conflict on depression score was greater among boys (adjusted coefficient beta = 1.076, 95% CI = 0.532 to 1.620, *p*-value < 0.001) than among girls (adjusted coefficient beta = 0.361, 95% CI = 0.093 to 0.629, *p*-value = 0.009). Table 3 shows the linear regression analysis, with depression score as the dependent variable. Figure 3 indicates multivariable normality since the residuals followed a normal distribution. Figure 4 indicates the homoscedasticity and linearity of the multivariable model, with depression score as the dependent variable.

### 3.6. Association Between TikTok Use and Sleepiness

Bivariate analysis showed that the mean sleepiness for girls was 5.91 (SD: 2.13), and for boys it was 5.61 (SD: 1.92) (*t*-test = 0.81, df = 217, mean difference = 0.30, 95% CI = −0.42 to 1.01, Cohen’s d = 0.15, *p*-value = 0.42). Pearson’s correlation coefficients between sleepiness and mood modification, conflict, and relapse were 0.35 (*p*-value < 0.001), 0.38 (*p*-value < 0.001), and 0.18 (*p*-value = 0.009), respectively.

We found that mood modification (adjusted coefficient beta = 0.655, 95% CI = 0.336 to 0.975, *p*-value < 0.001) and conflict (adjusted coefficient beta = 0.674, 95% CI = 0.379 to 0.969, *p*-value < 0.001) were associated with increased sleepiness in the full sample. Moreover, we identified a positive association between mood modification and sleepiness in both boys and girls. Also, stratified analysis showed that the impact of mood modification on sleepiness was higher among boys (adjusted coefficient beta = 1.030, 95% CI = 0.246 to 1.813, *p*-value = 0.011) than among girls (adjusted coefficient beta = 0.610, 95% CI = 0.255 to 0.965, *p*-value = 0.001). We found a statistically significant and positive association between conflict and sleepiness among girls (adjusted coefficient beta = 0.675, 95% CI = 0.319 to 1.031, *p*-value < 0.001), but this association did not exist among boys. Table 4 shows univariate and multivariable linear regression models with sleepiness as the dependent variable. Figure 5 indicates multivariable normality since the residuals followed a normal distribution. Figure 6 indicates the homoscedasticity and linearity of the multivariable model, with depression score as the dependent variable.

## 4. Discussion

We conducted a cross-sectional study to explore the association between TikTok use and anxiety, depression, and sleepiness in a sample of adolescents in Greece. We used a valid tool that specifically measures problematic TikTok use (i.e., the TikTok Addiction Scale), in contrast to previous studies that measured TikTok use with non-specific tools or even with simple questions that only measure the frequency of TikTok use [11,12,13,14]. The main findings of our study showed a positive association between TikTok use and anxiety, depression, and sleepiness in adolescents.

In our study, the mean time of daily TikTok use was 2.77 h. Data show that US adolescents spend 2.5 h a day on TikTok [1]. Also, Gentzler et al. found in a sample of American adolescents that the mean time of daily TikTok use is 2.17 h [12]. A recent study in China found a higher level of TikTok use, with a mean daily time of 2.85 h [11].

In our sample, the mean daily TikTok usage and mean TTAS score were statistically higher for girls than for boys. Also, we found that problematic TikTok use was higher among girls vs. boys. The literature supports our findings since Gentzler et al. found that the mean time of daily TikTok use was 2.80 h and 1.68 h for girls and boys, respectively, with this difference being statistically significant [12]. Moreover, girls spent more time on smartphones, social media, texting, and general computer use than boys, according to data from three large, representative surveys of adolescents in the US and the UK aged 13 to 18 [45]. Also, Kircaburun et al. employed a sample of university students in Turkey and found that women are more prone to being problematic social media users, meaning they have a harder time controlling how much time they spend on social media [46]. This difference between the two genders may be attributed to the fact that girls place greater importance on social relationships than boys do [47,48,49]. Given that social connections play a more significant role in the well-being of girls, their engagement with social media platforms may have a stronger link to their mental health. Additional factors at the intersection of gender and adolescent development could also be at play. For instance, social media might trigger feelings of upward social comparison [50], a process that could be particularly influential among female adolescents [51]. Furthermore, the heightened body image concerns experienced by girls may be intensified by their exposure to social media content [52]. This framework could be also an explanation for our finding that boys experience a stronger association between TikTok use and anxiety/depression than girls. Girls’ higher engagement with TikTok probably allows them to understand more deeply the way that this application may affect their mental health. In this context, girls may use TikTok in a more sophisticated way than boys by better understanding the possible risks of problematic TikTok use. In any case, these differences between the two genders should be further clarified by better-organized future studies. For instance, future studies may consider better methodological tools, like the measurement of anxiety and depression with psycho-physiological measures.

Our analysis revealed that the factors “mood modification” and “tolerance” had the highest average scores, followed by “conflict” and “salience”. The factors “relapse” and “withdrawal symptoms” showed the lowest mean scores. Further examination of the TTAS factor scores indicated that TikTok primarily affects adolescents’ mood modification and tolerance. Specifically, problematic TikTok users appear to increasingly rely on the platform to enhance their mood (i.e., salience). Consequently, this increased usage leads to decreased tolerance, as users require more TikTok engagement to feel satisfied. Conversely, the low average scores for “relapse” and “withdrawal symptoms” suggest that adolescents experienced minimal relapse and withdrawal effects. This implies that teenagers reported low levels of negative emotions when unable to use TikTok. Additionally, users infrequently reverted to previous TikTok usage patterns after periods of abstinence.

Moreover, we found a positive association between problematic TikTok use and anxiety symptoms. Two recent studies in China are in accordance with our findings, since scholars found that TikTok users with addictive behaviors show greater levels of anxiety [11,13]. The connection between problematic social media usage and psychological distress may be attributed to the fear of missing out [53]. Constant engagement with social platforms can provoke anxiety related to the fear of missing out, potentially leading to more frequent checking of these sites [54]. The effort to keep up with multiple social media accounts simultaneously can generate anxiety, which may intensify the fear of missing out and create a self-perpetuating cycle of increased stress [55]. Research has also indicated that excessive use of social media is associated with decreased creativity and a lowered perception of one’s intellectual abilities [56]. Spending considerable time on social media can reduce real-life peer interactions, potentially resulting in feelings of isolation, loneliness, and anxiety [53]. Additionally, individuals experiencing anxiety often prefer online communication due to difficulties with in-person interactions [57]. Problematic social media use may also increase worries about negative evaluations, enhance susceptibility to cyberbullying, and promote negative online exchanges, all of which can contribute to heightened levels of anxiety [58].

Our multivariable analysis identified a positive association between problematic TikTok use and depressive symptoms. Four studies in China, the US, and Serbia support this finding since they found that depressive symptoms are more frequent among problematic TikTok users [11,12,13,14]. Studies indicate that social media usage can have detrimental effects on emotional states, quality of life, and overall contentment [59,60,61]. For example, scrolling through social media platforms without actively interacting has been linked to reduced social bonds and heightened feelings of solitude and disconnection [62]. One potential reason for this is that viewing idealized representations of others’ lives on these platforms may evoke envy and the false belief that peers are experiencing more fulfilling or prosperous lives [63]. These emotions of jealousy can eventually lead to feelings of inadequacy and depression [64]. Another contributing factor might be the sense of squandering time on trivial social media activities, which could negatively influence one’s emotional state [61]. Moreover, the substantial rise in time dedicated to social media has led some experts to propose recognizing “internet addiction” as a distinct condition closely associated with depression [65,66]. An additional hypothesis suggests that depression may arise from reduced self-esteem when users make unfavorable comparisons between themselves and carefully selected images of seemingly more attractive, thinner, more popular, or more affluent individuals [67,68]. Finally, increased social media exposure may heighten the risk of online harassment, potentially contributing to symptoms of depression [69].

Moreover, we found that levels of sleepiness are higher among adolescents with greater levels of problematic TikTok use. Chao et al. arrived at a similar conclusion since they found that TikTok-addicted users exhibit poorer sleep [11]. Additionally, several other studies that investigated the association between social media use and sleep find that excessive social media use results in poor sleep quality, sleepiness during the daytime, and sleep disturbances [32,70,71,72,73,74,75].

There are several ways in which TikTok might be linked to mental health issues. First, cyberbullying on TikTok has emerged as a critical concern in recent times, drawing considerable attention. This online form of harassment and abuse significantly impacts the mental well-being of those targeted. On TikTok, cyberbullying manifests in various ways, including intimidation, verbal attacks, dissemination of false information, online stalking, and extortion. The most prevalent form occurs through user comments on uploaded content. While these comments can offer support and praise, they often contain hurtful, mean-spirited, or even sinister messages. Negative remarks about an individual’s appearance or behavior can be psychologically harmful, particularly for younger users who may already struggle with emotional and self-esteem issues [76,77,78]. Furthermore, the follower count that TikTok users accumulate activates their dopaminergic system, which is responsible for pleasure through the reward mechanism. This leads to addictive behaviors and a strong urge to repeat actions that initially triggered pleasure [79]. Moreover, extended exposure to blue light-emitting screens (such as those on smartphones, tablets, and computers) is linked to various sleep disorders, including insomnia, disrupted circadian rhythms, and sleeplessness [80,81]. A systematic review of studies involving school-aged children and adolescents revealed a negative correlation between nighttime screen use and sleep quality [82]. Additionally, another review identified a connection between social media usage and several sleep-related outcomes, including delayed bedtimes, altered sleep duration (early waking or sleep disturbances), daytime fatigue, sleep deficits, and overall sleep quality [71].

### Limitations

Our study had several limitations. First, we conducted a cross-sectional study; therefore, we cannot establish a causal relationship between TikTok use, anxiety, depression, and sleepiness. Thus, we cannot be sure whether problematic TikTok use increases anxiety, depression, and sleepiness or whether these mental health issues pre-exist and lead to increased TikTok use. Additionally, the short-term negative emotional dimensions detected during social media use could be missing in a cross-sectional study. In this context, the measurement of negative emotions such as frustration, indignation, and a sense of impotence, which is recognizable in a general sense of bitterness, could provide more evidence regarding the relationship between TikTok use and mental health issues [83]. Thus, longitudinal studies that explore the association between study variables could provide significant information. Second, our study population included adolescents aged 18 and 19 years. Also, we used a convenience sample; thus, our results cannot be representative of adolescent TikTok users. For instance, our study included mainly girls (81.3%); thus, this gender imbalance introduces selection bias. Future studies should include random or stratified samples of adolescents aged 10 to 19 years to produce more representative results. For instance, a more representative, randomized sample would improve the validity of the findings. Third, we used valid tools to measure TikTok use, anxiety, and depression. However, our participants may have compromised their answers due to social desirability bias. Therefore, information bias is probable in our study. Fourth, we measured sleepiness with a simple question. This single-item Likert scale introduced information bias in our study. Scholars conducting future studies should measure sleepiness in a more valid way by using established scales, such as the Epworth Sleepiness Scale [84], the Pittsburgh Sleep Quality Index [85], the Karolinska Sleepiness Scale [36], or the Stanford Sleepiness Scale [37]. Fifth, although we used a valid tool to measure TikTok use, other dimensions of TikTok use could also be measured. For instance, scholars in the future could measure the content displayed or the types of use performed by adolescents. Sixth, we eliminated only one confounder in our study. Since there are plenty of confounders in the relationship between TikTok use, anxiety, depression, and sleepiness, future studies should focus on eliminating these confounders. For instance, screen time, other social media use, social media habits, and sleep patterns should be considered as potential confounders in future studies. Seventh, the sample size for boys was small; thus, our multivariable models in the case of boys may lack sufficient power. Further studies with bigger samples could provide more reliable results in the stratification analysis. Finally, our linear regression models showed that TikTok use explains a small percentage of the variance in anxiety, depression, and sleepiness. Therefore, future research should also examine the impact of other predictors on anxiety, depression, and sleepiness. For instance, personality characteristics, cyberbullying, and the relationship between family and adolescents may affect anxiety, depression, and sleepiness or even interact with TikTok use. Moreover, the evaluation of potential mediating factors in the relationship between TikTok use and anxiety, depression, and sleepiness may further explain the impact of TikTok use. For instance, problematic TikTok use may negatively impact the relationship between adolescents and their families, thereby contributing to anxiety, depression, and sleepiness.

## 5. Conclusions

Our research revealed an association between TikTok usage and increased levels of anxiety, depression, and sleepiness among adolescents. Given the limitations of our study and the scarcity of research in this area, additional studies are required to establish more conclusive evidence regarding the relationship between TikTok use and anxiety, depression, and sleepiness. The early identification of TikTok-addicted users is essential to promote their mental health and well-being. Healthcare professionals should be alert to recognizing symptoms of problematic TikTok use among adolescents. Moreover, policymakers should develop and adopt appropriate interventions to reduce adolescents’ TikTok use.

Addressing mental health concerns in individuals who excessively use TikTok requires the creation of specific interventions. These may encompass awareness campaigns, programs promoting digital well-being, and the provision of counseling resources [11]. To effectively manage problematic TikTok usage and support adolescent mental health, it is essential for professionals, educators, and families to work together. Additionally, there is a need to establish guidelines and regulations that encourage healthy TikTok use. For instance, a recent systematic review examined the efficacy of psychological interventions delivered via mobile apps for college students, revealing that these interventions are well-received and adhered to [86]. Furthermore, initial evidence suggests that these interventions can be effective in treating various conditions, including stress, anxiety, depression, and risky behaviors such as alcohol and tobacco misuse, as well as in improving sexual knowledge.

## Figures and Tables

**Figure 1 pediatrrep-17-00034-f001:**
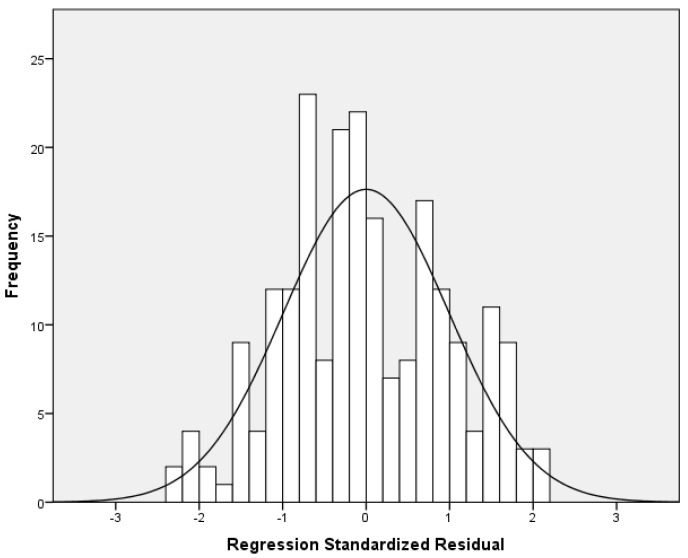
Histogram of the residuals with anxiety score as the dependent variable.

**Figure 2 pediatrrep-17-00034-f002:**
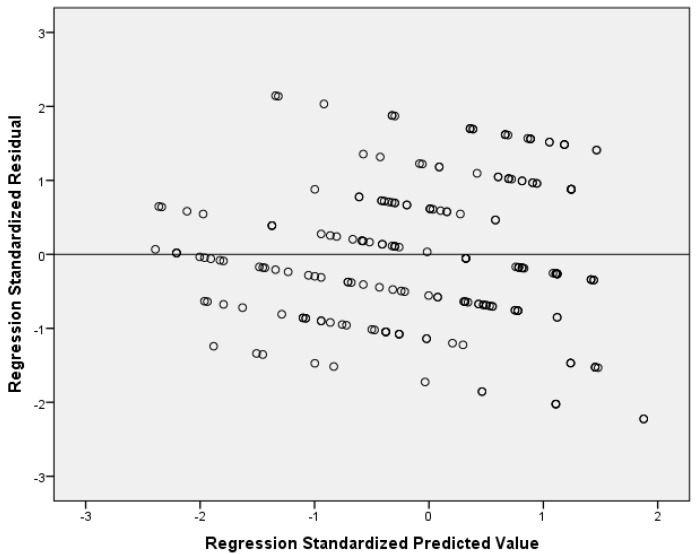
Scatterplot of residuals versus predicted values with anxiety score as the dependent variable.

**Figure 3 pediatrrep-17-00034-f003:**
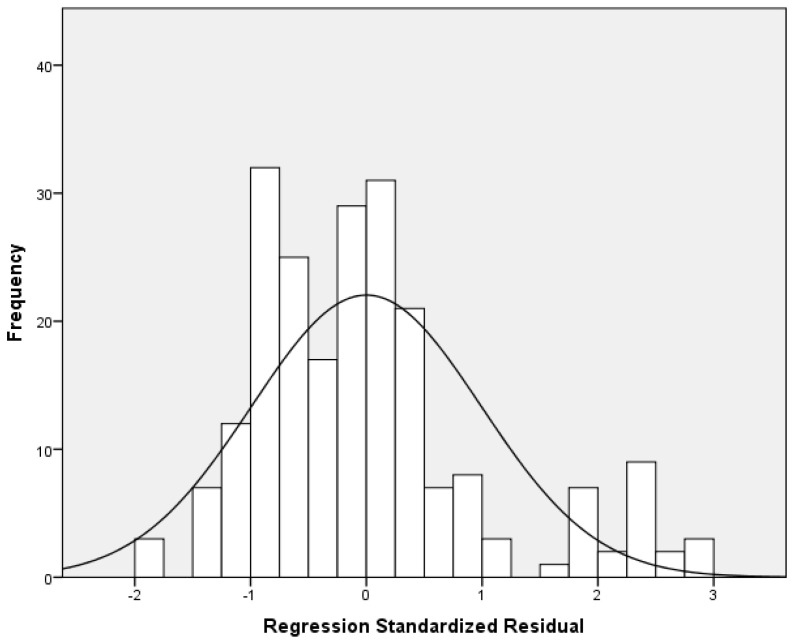
Histogram of the residuals with depression score as the dependent variable.

**Figure 4 pediatrrep-17-00034-f004:**
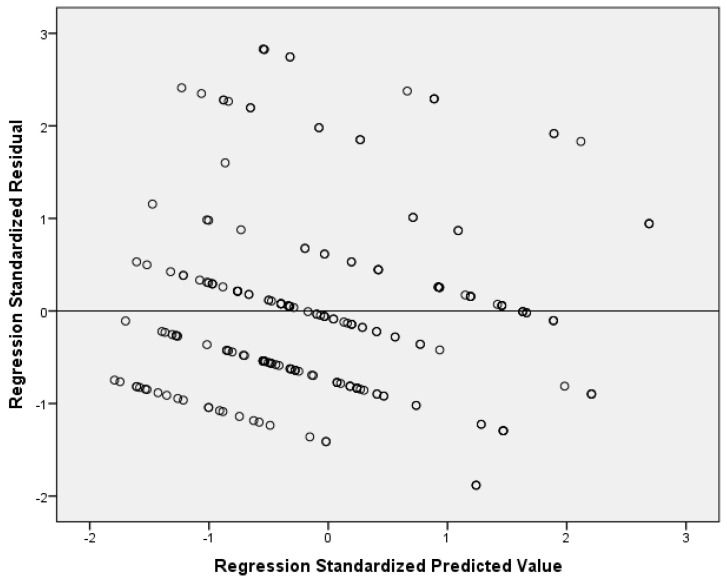
Scatterplot of residuals versus predicted values with depression score as the dependent variable.

**Figure 5 pediatrrep-17-00034-f005:**
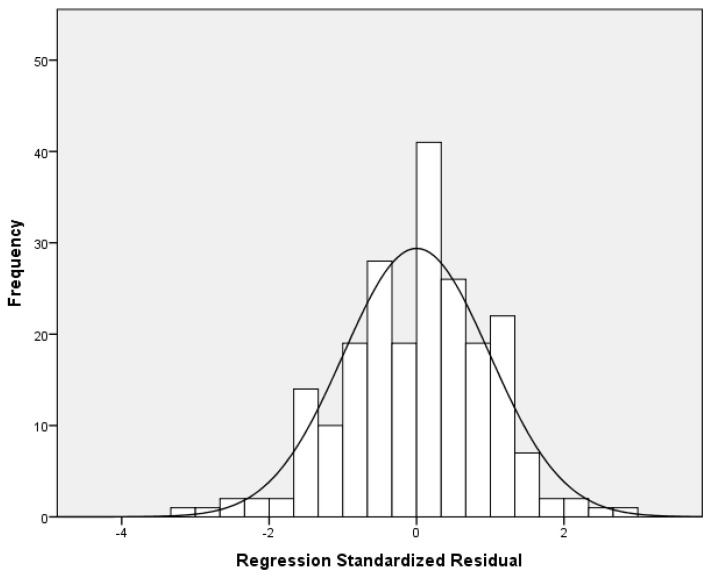
Histogram of the residuals with sleepiness as the dependent variable.

**Figure 6 pediatrrep-17-00034-f006:**
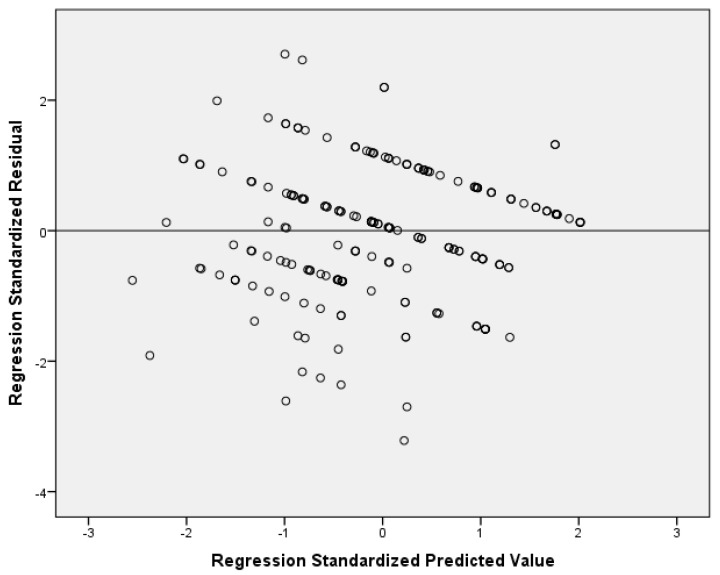
Scatterplot of residuals versus predicted values with sleepiness as the dependent variable.

**Table 1 pediatrrep-17-00034-t001:** Descriptive statistics for the TikTok Addiction Scale.

	Minimum Value	Maximum Value	Mean	Standard Deviation	Median	Skewness	Kurtosis
TikTok Addiction Scale	1.07	4.07	2.48	0.69	2.47	0.26	−0.55
Salience	1.00	4.50	1.99	0.85	2.00	0.67	0.06
Mood modification	1.00	5.00	3.64	0.89	3.50	−0.77	0.16
Tolerance	1.00	5.00	3.12	0.99	3.00	−0.03	−0.37
Withdrawal symptoms	1.00	3.00	1.45	0.66	1.00	1.19	0.15
Conflict	1.00	5.00	2.55	1.00	2.50	0.57	−0.36
Relapse	1.00	5.00	1.83	0.96	1.50	0.91	−0.01

**Table 2 pediatrrep-17-00034-t002:** Linear regression models with anxiety score as the dependent variable.

Independent Variables	Univariate Models			Multivariable Model ^a,b^			
	Unadjusted Coefficient Beta	95% CI for Beta	*p*-Value	Adjusted Coefficient Beta	95% CI for Beta	*p*-Value	VIF
Full sample ^b^							
Mood modification	0.476	0.224 to 0.728	<0.001	0.404	0.115 to 0.693	0.006	1.301
Conflict	0.266	0.034 to 0.497	0.025	0.098	−0.168 to 0.365	0.467	1.366
sRelapse	0.280	0.040 to 0.520	0.023	0.066	−0.220 to 0.351	0.651	1.450
Boys (n = 41) ^c^							
Mood modification	1.130	0.418 to 1.842	0.003	0.760	0.121 to 1.399	0.021	1.343
Conflict	1.392	0.926 to 1.858	<0.001	1.236	0.763 to 1.710	<0.001	1.126
Relapse	0.243	−0.524 to 1.010	0.525	−0.063	−0.660 to 0.533	0.831	1.264
Girls (n = 178) ^d^							
Mood modification	0.365	0.093 to 0.638	0.009	0.338	0.031 to 0.645	0.031	1.283
Conflict	0.019	−0.235 to 0.273	0.883	−0.273	−0.581 to 0.034	0.081	1.539
Relapse	0.283	0.031 to 0.534	0.028	0.304	−0.013 to 0.621	0.060	1.006

^a^ Multivariable models are adjusted for age. ^b^ R^2^ for the multivariable model = 4.8%, *p*-value for ANOVA = 0.006. ^c^ R^2^ for the multivariable model = 18.2%, *p*-value for ANOVA < 0.001. ^d^ R^2^ for the multivariable model = 4.1%, *p*-value for ANOVA = 0.023. CI: confidence interval, VIF: variance inflation factor.

**Table 3 pediatrrep-17-00034-t003:** Linear regression models with depression score as the dependent variable.

Independent Variables	Univariate Models			Multivariable Model ^a,b^			
	Unadjusted Coefficient Beta	95% CI for Beta	*p*-Value	Adjusted Coefficient Beta	95% CI for Beta	*p*-Value	VIF
Full sample ^b^							
Mood modification	0.296	0.065 to 0.527	0.012	0.051	−0.200 to 0.303	0.688	1.301
Conflict	0.542	0.344 to 0.740	<0.001	0.472	0.239 to 0.704	<0.001	1.366
Relapse	0.374	0.161 to 0.588	0.001	0.117	−0.132 to 0.366	0.355	1.450
Boys (n = 41) ^c^							
Mood modification	0.387	−0.358 to 1.131	0.300	−0.245	−0.979 to 0.489	0.503	1.343
Conflict	1.011	0.495 to 1.527	<0.001	1.076	0.532 to 1.620	<0.001	1.126
Relapse	0.538	−0.168 to 1.244	0.131	0.650	−0.035 to 1.336	0.062	1.264
Girls (n = 178) ^d^							
Mood modification	0.295	0.052 to 0.538	0.018	0.101	−0.166 to 0.369	0.456	1.283
Conflict	0.456	0.241 to 0.672	<0.001	0.361	0.093 to 0.629	0.009	1.539
Relapse	0.351	0.130 to 0.572	0.002	0.102	−0.174 to 0.379	0.466	1.620

^a^ Multivariable models are adjusted for age. ^b^ R^2^ for the multivariable model = 10.9%, *p*-value for ANOVA < 0.001. ^c^ R^2^ for the multivariable model = 28.2%, *p*-value for ANOVA = 0.003. ^d^ R^2^ for the multivariable model = 8.0%, *p*-value for ANOVA = 0.001. CI: confidence interval, VIF: variance inflation factor.

**Table 4 pediatrrep-17-00034-t004:** Linear regression models with sleepiness as the dependent variable.

Independent Variables	Univariate Models			Multivariable Model ^a,b^			
	Unadjusted Coefficient Beta	95% CI for Beta	*p*-Value	Adjusted Coefficient Beta	95% CI for Beta	*p*-Value	VIF
Full sample ^b^							
Mood modification	0.818	0.526 to 1.111	<0.001	0.655	0.336 to 0.975	<0.001	1.301
Conflict	0.789	0.529 to 1.049	<0.001	0.674	0.379 to 0.969	<0.001	1.366
Relapse	0.386	0.098 to 0.673	0.009	−0.217	−0.533 to 0.098	0.176	1.450
Boys (n = 41) ^c^							
Mood modification	0.981	0.254 to 1.708	0.009	1.030	0.246 to 1.813	0.011	1.343
Conflict	0.754	0.160 to 1.348	0.014	0.450	−0.130 to 1.031	0.124	1.126
Relapse	−0.117	−0.880 to 0.646	0.759	−0.621	−1.352 to 0.111	0.094	1.264
Girls (n = 178) ^d^							
Mood modification	0.790	0.462 to 1.118	<0.001	0.610	0.255 to 0.965	0.001	1.283
Conflict	0.794	0.498 to 1.089	<0.001	0.675	0.319 to 1.031	<0.001	1.539
Relapse	0.457	0.145 to 0.770	0.004	−0.178	−0.545 to 0.189	0.339	1.620

^a^ Multivariable models are adjusted for age. ^b^ R^2^ for the multivariable model = 19.0%, *p*-value for ANOVA < 0.001. ^c^ R^2^ for the multivariable model = 25.9%, *p*-value for ANOVA = 0.005. ^d^ R^2^ for the multivariable model = 18.4%, *p*-value for ANOVA < 0.001. CI: confidence interval, VIF: variance inflation factor.

## Data Availability

Our data are available from the following Figshare repository: https://doi.org/10.6084/m9.figshare.28311713.v1 (accessed on 3 March 2025).

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
