# Peer review of "Association Between TikTok Use and Anxiety, Depression, and Sleepiness Among Adolescents: A Cross-Sectional Study in Greece"

_pediatrrep, 2025, doi:10.3390/pediatric17020034_

Round 1

Reviewer 1 Report

Comments and Suggestions for Authors

The manuscript "Association between TikTok use and anxiety, depression, and sleepiness among adolescents: a cross-sectional study in Greece" describes the association between problematic TikTok use and some symptoms of well-being in a convenience sample of Greek adolescents. The strength of this study is the use of a newly developed inventory to assess problematic TikTok use. However, there are numerous methodological issues that prevent this manuscript from being published.

1. The authors claim on page 3 (lines 99-100) that they investigated the association between TikTok use and distress in a European country for the first time. Please note, however, that similar studies have been published previously, e.g.:

  • Rogowska, A.M.; Cincio, A. Procrastination Mediates the Relationship between Problematic TikTok Use and Depression among Young Adults. J. Clin. Med. 202413, 1247. https://doi.org/10.3390/jcm13051247

Therefore, “exploration” should not take place here. The authors should better search the literature on this topic and put forward directional hypotheses about the relationships between TikTok use and distress.

2. The article does not explain the concept of "sleepiness" at all. What are the authors measuring? A single-item and rather random question, without a theoretical basis and without the possibility of assessing the reliability of this single question, is unacceptable in science. Please remove "steepness" from the dependent variables and at most - after a comprehensive explanation in the introduction of what it even means - you can include this single-item measurement in the confounding or controlled variables.

3. The methodology section does not describe how the time of TikTok use was characterized and in what time frame (e.g. last day, typically in the last week? or otherwise?). Please provide the exact content of the question and answer options.

4. There are more assumptions about the use of linear regression, beyond the absence of multicollinearity, including linearity, homoscedasticity of residuals, multivariate normality, and lack of endogeneity. Please report the results of tests for all of the criteria listed above, regardless of whether they are met.

5. the study participants did not differ in age in principle - they were all 18 or 19 years old, as the description suggests. It is not understandable or theoretically justified why the authors decided to adjust for age?

6. Gender should be recoded 0-1 (e.g. 0 = boys, 1 = girls) and included in one common regression model. There is no justification for conducting a separate model for 40 boys. What is the strength of the effects in this case? What is the power of such a test? What is the point? Similarly, conducting separate models for the extra-short 2-item scale of depression and anxiety symptoms is unjustified. The authors should conduct a single analysis for distress symptoms for the composite PHQ-4 score. Gender recoded 0-1 and at most "sleepness" should be included in the model (but only and exclusively after a thorough explanation of what this even means). This analysis should be preceded by an analysis of differences between the sexes using the Student's t-test or the alternative nonparametric Mann-Whitney U test (depending on meeting the initial conditions). The regression analysis should also be preceded by a Pearson or Spearman correlation analysis (as above, depending on the criteria).

7. Please provide the effect size for the Student's t-test (e.g. Cohen's d), report and interpret the results in the manusctript.

8. Please improve the results of prevalence comparison (page 4, line 190-191). What test was used for this purpose? If you used Pearson's test of independence, please report all results, chi-square statistics, df, p, as well as effect size (e.g. phi or V).

9. Table 1 should include skewness and kurtosis.

10. What does it mean "Mean anxiety score was 2.95 (SD; 1.74) with a median value of 3 hours
(range; 0 to 6), while mean depression score was 2.11 (SD; 1.57) with a median value of 2 hours (range; 0 to 6)"? How did you assess anxiety in hours?

11. It is unclear what mean "mood modification" and how it was assessed n the study. Could you explain or remove it?

12. The manuscript requires total change, also in the discussion, since there will be different results of the analyses than currently.

Author Response

Dear Reviewer,

Reviewer 2 Report

Comments and Suggestions for Authors

Abstract: 

  1. "We employed a convenience sample of adolescents in Greece." - we need more information about the sample: age range, Mean, SD, gender ration, place (TikTok?) and locale of the participants.
  2. Line 23:  "Our multivariable analysis ... " what does it mean? Multivariate analysis? Which one? Please clarify.
  3. Among the results,give some concrete findings (e.g., betas, p. values etc).
  4. Line 29: Instead of mentioning limitations, a more concrete conclusion, e.g., suggestions should be done.

Introduction

  1. Line 38: over 1 billion users worldwide - this "fact" needs a reference.
  2. There are more studies on the mental health issues related to TikTok please look after, e.g., https://www.sciencedirect.com/search?qs=TikTok%20use%20and%20mental%20health Also, there are more papers on the issue, see. e.g., Yao et al., 2023 https://doi.org/10.1016/j.chb.2023.107751
  3. Also, there is an ample evidence and rational to justify, why the authors think these associations need to be further explored?
  4. Lines 91-103. Please complete this paragraph with giving the concrete research questions with hypotheses with references

Sample

  1. Even though it is a convenience sample, more information is need about their characteristics, inclusion criteria and particularly the ethical issues (voluntary or not, anonymous or not, how about informed consent and also from the parents due to  < 18 age of the participants). "We informed our participants about the 148
    study design, and we asked whether they consent to participate in our study. In 149" that way, we obtained informed consent.
    " - how was this process going on? Online?

Methods:

  1. Were the Greek version of the scales validated? If yes, give a reference to them.

Results:

  1. Table 1. I suggest to add minimum and maximum values first, then Mean (SD), and also to be completed with skewness and kurtosis. For all scales, also for mental health scales.
  2. How about gender differences? Since usually there are gender differences in mental health issues in adolescence, it can be relevant here.
  3. Table 2: number of boys and girls should be given.
  4. How about VIF statistics for multivariate analyses? To avoid multicollinerarity, these should be checked.

Discussion and conclusions

  1. Apporpriate. One remark: are there any interventions to give an outine to TikTok use to youth?

Author Response

Dear Reviewer,

Reviewer 3 Report

Comments and Suggestions for Authors

This study explores the association between TikTok use and mental health indicators—anxiety, depression, and sleepiness—among adolescents in Greece. Given the increasing popularity of TikTok, particularly among younger users, investigating its potential negative psychological effects is highly relevant. However, while the study provides important preliminary insights, there are areas where it could be refined for greater clarity, methodological rigor, and impact.

Starting from the unilateral view of technology that should be better enlarging by positive one (see for instance Greitemeyer, T. (2022). Prosocial modeling: Person role models and the media. Current Opinion in Psychology, 44, 135-139. and D’Errico, et al. Scare-away risks: the effects of a serious game on adolescents’ awareness of health and security risks in an Italian sample. Multimodal Technologies and Interaction).

The study uses a cross-sectional design, which limits the ability to infer causality. It remains unclear whether TikTok use contributes to anxiety and depression or whether pre-existing mental health issues lead to increased TikTok consumption. A longitudinal study would provide stronger evidence. Furthermore, the short-term negative emotional dimension detected during interactions with social media would be missing, and in this sense negative emotions such as indignation, frustration, sense of impotence, recognizable in a general sense of bitterness could provide more elements of understanding of this study (see for example Poggi, & D’Errico (2010). The mental ingredients of bitterness. Jmui).

Furthermore, the contents displayed, what type of use is made, are all levels that should be improved in future studies, and recognized at this stage as limitations.

From a methodological point of view, the study relies on a convenience sample, which may introduce selection bias and limit generalizability beyond the specific adolescent population in Greece. A more representative, randomized sample would improve validity.

The results suggest that boys experience a stronger association between TikTok use and anxiety/depression than girls, yet girls report higher TTAS scores and daily TikTok use. A deeper discussion of these findings would help clarify why boys may be more vulnerable despite lower engagement. For this purpose, future studies should also consider other methodological tools, like the detection of anxiety and engagement in psycho-physiological measures.

While the study focuses on TikTok use, other lifestyle factors (e.g., screen time, social media habits, sleep patterns) could also influence anxiety, depression, and sleepiness. Including these in the analysis could enhance explanatory power.

The paper does not sufficiently explore why TikTok might be linked to mental health problems. Potential mechanisms (e.g., social comparison, cyberbullying, sleep disruption, or dopamine-driven addiction patterns) should be discussed.

This study makes a good contribution to understanding the potential risks of TikTok use among adolescents. However, refining the methodological approach, expanding the discussion of mechanisms, and addressing limitations in literature, and other methodological choices would enhance its impact and validity. Future research should adopt longitudinal designs and broader samples to strengthen causal inferences and provide more actionable recommendations.

Author Response

Dear Reviewer,

Reviewer 4 Report

Comments and Suggestions for Authors

After reading your manuscript several concerns raised regarding methodology and fit of the article in the journal scope.

Literature review is insufficiente to clearly explain the importance and significance of the study in the field of pediatrics. This study could be more appropiate in a psychology or psychiatric journal considering the variables examined and the age of the participants. What was the novelty of the study beyond analysing TikTok use

Even when a study is exploratory, the introduction should end with clearly articulate hypothesis describing the expected results.

Authors mention that participants were recruited via TikTok and convenience sampling, but they do not elaborate on how the recruitment process was conducted: steps taken to recruit participants, whether there were any measures taken to avoid selection bias, response rate and how many individuals were invited versus how many completed the survey, inclusion of manupulation check to avoid random responses, etc.

An importante issue is that the sample was formed predominantly by girls (81.3%), which in my opinion is skewing the results. Gender imbalance is not even mention in the limitations section. At least, the sample should be 60%-40%. I would also like to know how gender was assessed. Non-binary individuals were not included in the study or authors asked about sex assigned at birth?

Authors to explicitly state the methods they used to check for normality in the data distribution.

Authors should explain how the cut-off point to determine problematic TikTok use was establish. Was it based on previous research?

REsults showed that only a small percentage of the variance in anxiety, depression, and sleepiness is explained by problematic TikTok use. While the findings are still valuable, authors need to explain that their models explain only a small portion of the variance and discuss what other factors may be more crucial than TikTok use or maybe discussion that mediation models with other variables could be a more valuable contribution in the future.

Author Response

Dear Reviewer,

Round 2

Reviewer 1 Report

Comments and Suggestions for Authors

Although I disagree with the arguments in many places, I believe that the authors have put a great deal of effort into improving the manuscript and, subject to numerous limitations, I agree with its publication in its current version.

Reviewer 2 Report

Comments and Suggestions for Authors

The authors did a great job in replying all the reviewer requests carefully. I suggest publishing in this current improved version.

Reviewer 3 Report

Comments and Suggestions for Authors

The paper in the present version is ready for publication, the authors have been very thorough in their answers and in the overall revision of the paper. Congrats!